# Vanadium Pentoxide Exposure Causes Strain-Dependent Changes in Mitochondrial DNA Heteroplasmy, Copy Number, and Lesions, but Not Nuclear DNA Lesions

**DOI:** 10.3390/ijms241914507

**Published:** 2023-09-25

**Authors:** Nick L. Dobson, Steven R. Kleeberger, Adam B. Burkholder, Dianne M. Walters, Wesley Gladwell, Kevin Gerrish, Heather L. Vellers

**Affiliations:** 1Health and Exercise Department, University of Oklahoma, Norman, OK 73019, USA; lepageou39@ou.edu; 2National Institute of Environmental Health Sciences, Research Triangle Park, NC 27709, USA; kleeber1@niehs.nih.gov (S.R.K.); adam.burkholder@nih.gov (A.B.B.); gladwell@niehs.nih.gov (W.G.); gerrish@niehs.nih.gov (K.G.); 3Department of Physiology, Brody School of Medicine, East Carolina University, Greenville, NC 27834, USA; waltersd@ecu.edu; 4Department of Kinesiology and Sport Management, Texas Tech University, Lubbock, TX 79409, USA

**Keywords:** mitochondrial sequencing, mtDNA copy number, mtDNA damage, heteroplasmy, vanadium pentoxide

## Abstract

Interstitial lung diseases (ILDs) are lethal lung diseases characterized by pulmonary inflammation and progressive lung interstitial scarring. We previously developed a mouse model of ILD using vanadium pentoxide (V_2_O_5_) and identified several gene candidates on chromosome 4 associated with pulmonary fibrosis. While these data indicated a significant genetic contribution to ILD susceptibility, they did not include any potential associations and interactions with the mitochondrial genome that might influence disease risk. To conduct this pilot work, we selected the two divergent strains we previously categorized as V_2_O_5_-resistant C57BL6J (B6) and -responsive DBA/2J (D2) and compared their mitochondrial genome characteristics, including DNA variants, heteroplasmy, lesions, and copy numbers at 14- and 112-days post-exposure. While we did not find changes in the mitochondrial genome at 14 days post-exposure, at 112 days, we found that the responsive D2 strain exhibited significantly fewer mtDNA copies and more lesions than control animals. Alongside these findings, mtDNA heteroplasmy frequency decreased. These data suggest that mice previously shown to exhibit increased susceptibility to pulmonary fibrosis and inflammation sustain damage to the mitochondrial genome that is evident at 112 days post-V_2_O_5_ exposure.

## 1. Introduction

Interstitial lung diseases (ILDs) are lethal lung diseases characterized by pulmonary inflammation and progressive lung interstitial scarring [1]. Most ILDs result in interstitial fibrosis, the most common of which is Idiopathic Pulmonary Fibrosis, or IPF. The etiology of IPF is poorly understood, and pulmonary fibrosis is a complex trait that develops due to genetic and environmental factors [2,3]. A comprehensive review by Moss et al. [4] describes the current knowledge on biological and genetic contributions to IPF susceptibility and development. While our understanding of the underlying pathways(s) has progressed, we still have yet to determine all of the factors that holistically contribute to the development of IPF. Significantly, an individual’s genetic background can influence their susceptibility to IPF. To determine the contribution of genetics to a phenotype, we must assess the entire genome—consisting of both the nuclear and mitochondrial—for their association and interaction with that trait. To date, the role of the mitochondrial genome in IPF susceptibility remains unknown. 

Briefly, while the nuclear genome composes most cellular DNA (~99%), the mitochondrial genome is critical in regulating mitochondrial function. Mitochondrial function is partially dependent on the interaction of coding instructions from nuclear DNA (nDNA) and its DNA (mitochondrial DNA, mtDNA) [5,6]. Approximately 1500 nuclear genes encode mitochondrial proteins (10); however, 37 genes in the mitochondrial genome encode components of the oxidative phosphorylation (OXPHOS) multi-subunit complexes (Complexes 1, 3, 4, and 5 only), 13 of which are protein-coding (11). Furthermore, mtDNA can exist from ten to thousands of copies in each cell. Across these mtDNA copies, sequence changes can occur, and when these sequence changes occur in more than approximately 2% of the cellular population, it is termed heteroplasmy. Though poorly understood, mtDNA heteroplasmy has been linked to multiple diseases with a metabolic origin [7]; most data indicate that an alternate allele frequency of roughly 60–80% must be reached before cellular detriments occur [8]. Moreover, mitochondrial DNA (mtDNA) is more susceptible to damage and possible mutations than nuclear DNA due to its proximity to the electron transport chain and lack of histone protection [9]. Moreover, because the mitochondrial and nuclear genomes are codependent, there may be associations and interactions between these genomes that contribute to IPF onset. Based on its dynamic nature and susceptibility to oxidative damage, mtDNA is also likely vulnerable to the effects of excessive oxidative damage caused by environmental particulate matter that has entered the pulmonary system. Thus, ensuing alterations to mitochondrial function could lead to a vicious inflammatory cycle linked to biological pathways of IPF [4].

To study the pathogenesis of fibrogenesis in ILDs, our group previously developed an inbred mouse model of ILD in which we used vanadium pentoxide (V_2_O_5_), a common component of cigarette smoke, fuel ash, mineral ores, and steel alloys [10]. Using our previously established ILD mouse model in 49 BXD-RI strains (C57BL/6J and DBA/2J cross), we identified significant gene candidates on chromosome 4 associated with fibrosis across the strains (a list of gene candidates is discussed in Walters et al. work; [10]). The DBA/2J (D2) strain, which presented substantially higher pulmonary fibrosis and inflammation than the C57BL/6J (B6) strain following V_2_O_5_ exposure, had more genes associated with IPF development than the B6 strain (2). While these data indicated a significant genetic contribution to ILD susceptibility, they did not include any potential associations and interactions with the mitochondrial genome that might influence disease risk. Chronic exposure to V_2_O_5_ has been shown to increase reactive oxygen species (ROS) production [11,12,13,14] and induce mitochondrial dysfunction [15]. Thus, mtDNA mutations could arise from chronic V_2_O_5_ exposure and potentially compromise mitochondrial function and contribute to fibrosis and inflammation. However, the association between the mitochondrial genome and IPF susceptibility following V_2_O_5_ exposure still needs to be investigated.

Our primary goal of this pilot investigation was to assess the role of the mitochondrial genome with IPF susceptibility in the inbred strains we previously classified as the V_2_O_5_-resistant B6 and V_2_O_5_-responsive D2 strains. To conduct this project, we used ultra-deep sequencing to measure the complete lung mitochondrial mtDNA, and we assessed lesions, copy numbers, and heteroplasmy to determine if these characteristics contributed to IPF susceptibility and/or changed due to exposure. We then compared these factors at 14- and 112-days post-V_2_O_5_ exposure. We hypothesized that these strains would have unique mtDNA variants that could account for these differential responses and that the strains that presented high pulmonary inflammatory and fibrosing responses from V_2_O_5_ would show significant mtDNA damage and decreased heteroplasmy.

## 2. Results

### 2.1. mtDNA Variants and Heteroplasmy Frequency

Consistent with our previous work [10], we used B6 mice (*n* = 14) as the reference strain to identify mtDNA variants in the D2 mice (*n* = 13). Importantly, both B6 and D2 mice are the progenitors of numerous BXD recombinant inbred strains, and differences found between these strains can provide justification for additional investigations with a more diverse battery of inbred mice [16]. Using ultra-deep sequencing, we identified one unique mtDNA variant in the D2 mice (*n* = 13) in the *mt*-*Nd3* (mitochondrially encoded NADH dehydrogenase; T9461C) gene. Because mtDNA is maternally inherited, and variants are typically the result of adaptations to dietary and environmental factors throughout generations [17,18], it is unlikely that major mtDNA sequence alterations (i.e., 80–100% alternate allele frequency) will occur in response to acute exposure to particulate matter. Thus, this mtDNA variant in the D2 strain did not result from V_2_O_5_ exposure. 

Unlike alterations to the major mtDNA sequence (i.e., mtDNA variants), mitochondrial heteroplasmy can occur following brief exposures to oxidative stress-inducing agents such as excessive oxygen [19]. Therefore, strain-dependent changes in heteroplasmy frequency could signal variable adaptive responses to a given stimulus. For this investigation, we defined a position with heteroplasmy by the following conditions: A position with at least 1000 read sequencing depth where the alternate allele had one percent or higher occurrence than the reference allele. The one percent threshold was selected because ultra-deep sequencing can be prone to errors at very low heteroplasmy levels. For mtDNA lung heteroplasmy frequency, the results of a three-way ANOVA indicated an insignificant effect of time (*p* = 0.43), strain (*p* = 0.71), and treatment (*p* = 0.76) (Figure 1). However, because mtDNA heteroplasmy is highly variable [7], and techniques (laboratory and bioinformatic) to control and understand this variability are lacking, we feel it is essential to highlight a few noteworthy trends in our data: At 14 days post-exposure, the PBS-exposed B6 mice had a combined heteroplasmy frequency of 1% (±1.0%). The V_2_O_5_-exposed animals had a 0.3% (±0.3%) heteroplasmy frequency (Figure 1). In comparison, D2 PBS-exposed mice showed a combined heteroplasmy frequency of 6% (±3.0%) versus 4% (±4.0%) in the V_2_O_5_-exposed animals (Figure 1). 

At 112 days post-PBS exposure, B6 animals showed no changes in combined heteroplasmy frequency. However, the V_2_O_5_-exposed B6 animals had a combined heteroplasmy frequency of 9.5% (±5.6%) (Figure 1). In comparison, the PBS-exposed D2 mice showed a combined heteroplasmy frequency of 15% (±5.0%), whereas the V_2_O_5_-exposed mice decreased to 2% (±1.0%) heteroplasmy (Figure 1). These results indicate that acute exposure to V_2_O_5_ and PBS elicits adaptive responses that are strain-dependent. 

Lastly, we identified a few notable heteroplasmic positions in one or more D2 and B6 animals at each exposure or time point. First, as indicated in Table 1, 4 of the 13 heteroplasmic positions were denoted as nonsynonymous and within a gene encoding for oxidative phosphorylation. Further, the heteroplasmy frequency among those listed in Table 1 ranged from 1.1% to 26.0%. While the heteroplasmic sites were variable, they could be essential locations within the mitochondrial genome that eventually incur mutations over time and/or with continual V_2_O_5_ exposure.

### 2.2. DNA Damage and mtDNA Copy Numbers

Thousands of mtDNA copies exist in a given cell, and while not a perfect representation of overall mitochondrial function, changes in the mtDNA copy number can reflect alterations in energy production in response to various stimuli [20]. Using the protocols from Santos [21] and Furda et al. [22], we quantified DNA lesions (mitochondrial and nuclear) and mtDNA copy numbers in the lung tissue of both B6 and D2 strains following either PBS or V_2_O_5_ exposure. 

For the mtDNA copy number, the results of a three-way ANOVA did not yield a significant main effect (*p* = 0.12), but we did find a significant interaction between treatment and strain, where, overall, the V_2_O_5_-exposed D2 animals had significantly less lung mtDNA copies than D2 control animals (Main effect: PBS-exposed D2 mice = 43,159 ± 2903 mtDNA copies versus V_2_O_5_-exposed D2 mice = 7064 ± 5554 mtDNA copies, *p* = 0.02; Figure 2). This suggested that the decrease in mtDNA copy number in D2 mice could be the result of mitophagy due to oxidative stress-induced damage or alterations to mtDNA replication as a result of V_2_O_5_ exposure.

To complement our assessment of mtDNA copy numbers, we also measured lesions. DNA lesions represent structural damage to DNA that can include single-strand breaks, double-strand breaks, and mismatches, among others. We did not find a main effect (*p* = 0.07). Still, we found a significant interaction for treatment, time, and strain, where the V_2_O_5_-exposed D2 animals acquired significantly more mtDNA lesions at 112 days than their control counterparts at the same time point (112 days post-exposure: PBS-exposed D2 mice = 0.64 ± 0.23 mtDNA lesions/10Kb versus V_2_O_5_-exposed D2 mice = 2.82 ± 1.53 mtDNA lesions/10Kb, *p* = 0.04; Figure 3). Lastly, we assessed nDNA lesions, but we found no statistically significant differences between groups (*p* = 0.53), and no main effects or interactions (Figure 4). This suggested that mtDNA is more prone to damage than nDNA in response to acute V_2_O_5_ exposure.

## 3. Discussion

Our group [10] and others [2,3] have previously shown that an organism’s genotype can influence IPF susceptibility following exposure to V_2_O_5_ and other known environmental toxicants and particulate matter. By employing our established V_2_O_5_ exposure model in a battery of the BXD-RI strains (B6 and D2 cross) [10], our group previously identified a significant QTL on chromosome 4 for lung fibrosis (V_2_O_5_-induced collagen phenotype). Additionally, we identified several considerable gene candidates, and we found that most alleles linked to the collagen phenotype were from the D2 strain. From this work, we found that the D2 animals acquired significantly more pulmonary fibrosis and inflammation than the B6 animals; thus, we classified these animals as V_2_O_5_-responsive and -resistant strains, respectively. While these findings and others have provided further support linking genetics with IPF susceptibility, they only offered us information from the nuclear genome. As discussed in the introduction section, to assess the genetic contribution to a given trait, we must consider the nuclear and mitochondrial genome associations and interactions with that trait, and our understanding of these components is a critical piece in evaluating IPF susceptibility.

To conduct this preliminary investigation, we used our previously classified V_2_O_5_-resistant B6 and V_2_O_5_-responsive D2 strains and employed our established V_2_O_5_ mouse exposure model [10] to compare mitochondrial genome characteristics at 14- and 112-days post-exposure. Overall, a significant finding of this work was that the V_2_O_5_-responsive D2 animals had significantly more mtDNA lesions and fewer mtDNA copies than the V_2_O_5_-resistant B6 animals at 112 days post-exposure. We also found an interesting trend in D2 animals: Heteroplasmy frequency was lower than that of their control counterparts. Thus, our result shows that V_2_O_5_ exposure significantly compromises the mitochondrial genome, occurring 112 days post-exposure. This is the first study to assess mitochondrial genomic contributions to IPF susceptibility.

To assess the complete lung mitochondrial genomes in the V_2_O_5_-resistant B6 and V_2_O_5_-responsive D2 animals, we used ultra-deep sequencing at 14- and 112-days post-exposure. We used the B6 mouse as the reference, which is the standard. From this, we identified a unique mtDNA variant in the D2 mice located in the *mt-Nd3* (T9461C) mitochondrial gene (Table 1). However, this variant was not a result of V_2_O_5_ exposure. Even though we predicted possible changes in heteroplasmy—changes in sequence across mtDNA copies—we did not expect V_2_O_5_ to induce variants since they tend to develop over generations rather than in response to acute environmental exposures [17,18]. While we cannot speculate about the specific role of this unique mtDNA variant in the V_2_O_5_-induced pulmonary fibrosis phenotype of D2 mice, future investigations could potentially identify such a role.

We next asked if there were inherent or V_2_O_5_-induced effects on heteroplasmy frequency in the B6 and D2 animals. While we hypothesized that decreases in heteroplasmy would correspond to the fibrosing and inflammatory phenotype, we did not find statistically significant differences in heteroplasmy frequency on the basis of strain, time, or exposure (Figure 1). However, since heteroplasmy is dynamic in nature and highly variable within and across cells, our finding was not entirely surprising. Notwithstanding, we did find a noteworthy trend: At 112 days post-exposure, the V_2_O_5_-exposed D2 animals had a lower heteroplasmy frequency than their control counterparts at the same time point (2% versus 15%, respectively) (Figure 1). This trend is noteworthy for two reasons: (1) This trend aligns with the higher mtDNA lesions and lower copy numbers we observed at 112 days post-exposure in V_2_O_5_-exposed D2 mice (described below) and (2) on the basis of our previously published work [23], we have speculated that greater heteroplasmy frequency below the biochemical disease threshold (i.e., less than a 60–80% alternate allele frequency) could promote heightened responsiveness to various stimuli due to the greater genetic diversity of the organism. In this case, V_2_O_5_ exposure could lead to changes to the mitochondrial genome—and corresponding changes to mitochondrially encoded proteins within the respiratory chain complexes—that minimize ROS production following subsequent V_2_O_5_ exposure. Alternatively, a decrease in heteroplasmy frequency would suggest a blunted adaptive response to V_2_O_5_ exposure characterized by increased ROS production and accompanying increases in fibrotic and inflammatory signaling. When taken together, the fibrotic and inflammatory phenotype in D2 mice that we observed in our previous study corresponds to a decrease in heteroplasmy in this current work, suggesting a potential strain-dependent response to V_2_O_5_ exposure and an increased susceptibility to IPF in D2 mice. While our inability to demonstrate a statistically significant difference in heteroplasmy frequency was likely influenced by the low sample sizes used in some groups and the known variability of heteroplasmy, we believe that these trends provide essential information for investigating the role of an organism’s inherent ability to generate beneficial heteroplasmy.

We further assessed the specific heteroplasmic locations in the mitochondrial genomes of the animals in this study, presented in Table 1. Again, while heteroplasmy frequency and location were variable, they are worth noting because the understanding of heteroplasmy is ongoing. Table 1 provides a list of the heteroplasmic loci we identified in the animals of this study. One notable heteroplasmic position occurred at 112 days post-exposure: V_2_O_5_-exposed D2 animals showed a combined heteroplasmy frequency of 2.7% in the alternate allele of a mitochondrially encoded tRNA valine (*mt-Tv*) gene at location G1054A. Similarly, at 112 days post-exposure, the V_2_O_5_-exposed B6 mice had a combined heteroplasmy frequency of 26.0% in the mitochondrially encoded NADH dehydrogenase (*mt-nd3*) gene (T9461C). Interestingly, D2 animals also showed a combined heteroplasmy frequency of 20.2% in the mitochondrially encoded NADH dehydrogenase (*mt-nd5*; T11879C) gene 112 days post-PBS exposure. Together, these findings suggest three possible things: (1) V_2_O_5_ causes severe detriments to critical sites within the mitochondrial genome that are responsible for assembling proteins that carry out oxidative phosphorylation; (2) the effects of V_2_O_5_ on the mitochondrial genome do not occur immediately but become present by four weeks post-exposure in mice (in humans, this might be 20 or more years); (3) substantial heteroplasmy can still occur in the absence of noxious environmental stimuli in a strain-dependent manner. While the overall impact of these V_2_O_5_-induced heteroplasmic sites on oxidative phosphorylation is not yet known, it should be a key area for future work since V_2_O_5_ exposure has been linked to mitochondrial dysfunction and damage. Additionally, a complicated gene–environment dynamic appears to mediate the development of heteroplasmy, as substantial heteroplasmy occurred with or without exposure to V_2_O_5_. These findings are difficult to reconcile, and more extensive investigations are likely needed to better understand heteroplasmy and its role in V_2_O_5_-induced IPF.

Given its proximity to the respiratory chain complexes [24], mtDNA is frequently subjected to oxidative stress, and excessive oxygen can damage mtDNA and sometimes lead to mutations [15]. In this scenario, changes to the mitochondrial genome due to ROS exposure can result in a mixture of mutant and wild-type alleles across mitochondrial copies (i.e., mitochondrial heteroplasmy) [25]. Interestingly, the dynamic interplay between nuclear and mitochondrial genomes can also influence an organism’s phenotype [26]. Using cultured cybrid cell lines harboring a mitochondrial tRNA^Leu(UUR)^ mutation (A3243G), Kopinski et al. [26] showed that heteroplasmy leads to diminished acetyl coA generation and concomitant alterations in histone acetylation (i.e., retrograde signaling; mitochondria to the nucleus) [27]. Heteroplasmy frequency has also been shown to play a role in the corresponding phenotype in humans, with low levels (20–30%) in the mitochondrial tRNA^Leu(UUR)^ gene (A3243G) leading to the development of type 2 diabetes and high levels (80–90%) leading to deadly perinatal diseases [7]. Although we identified two instances of heteroplasmy at or exceeding 20% in B6 (*mt*-*Nd3* gene; T9461C) mice at 112 days post-V_2_O_5_ exposure and D2 (*mt*-*Nd5* gene; T11879C) mice at 112 days post-PBS exposure, future work is required to determine whether these heteroplasmic sites and/or others alter mitochondrial function and/or contribute to differential phenotypes in inflammation and fibrosis following V_2_O_5_ exposure. 

Finally, we assessed mtDNA lesions, copies, and nDNA lesions. We did not find any differences between strains at 14 days post-V_2_O_5_ exposure. However, D2 animals exposed to V_2_O_5_ had more mtDNA lesions (Figure 3) and fewer copy numbers (Figure 2) than D2 control animals 112 days post-exposure. To date, this is the first investigation that has assessed the effect of V_2_O_5_ on lung mitochondrial genomic characteristics. While it is beyond the scope of this work to detail potential functional and/or structural detriments resulting from damaged mtDNA, mutations can occur from damage [28]. mtDNA mutations can lead to disease and impair mitochondrial function; thus, V_2_O_5_-induced mtDNA lesions could result in mitochondrial dysfunction in affected pulmonary cells and exacerbate oxidative stress, leading to a vicious cycle of inflammation that is a critical factor linked to IPF development. Importantly, our investigation only considers acute exposure, and long-term exposure to V_2_O_5_ or similar compounds would likely result in more severe functional and structural impairments in the pulmonary system and other systems. mtDNA lesions may provide an important molecular target and/or a biomarker for IPF risk or progression. 

In addition to their increased mtDNA lesions, D2 V_2_O_5_-exposed animals also had reduced mtDNA copies at 112 days post-exposure, which is supported by other findings [29]. For example, Hong et al. [30] conducted a large-scale study involving 45,665 whole blood samples from participants derived from the U.K. Biobank and found a linear inverse relationship between carbon and nitric oxide (NO_2_) concentrations and mtDNA copy number. Similarly, using whole blood from 60 truck drivers and 60 office workers in Beijing before the 2008 Summer Olympic Games, Hou et al. [31] showed an inverse relationship between mtDNA copy number and exposure to personal particulate matter and elemental carbon, with more significant exposures corresponding to decreases in mtDNA copy number. As such, our findings align with other large-scale investigations that show an increase in mitochondrial genome impairments in response to environmental factors known to cause oxidative stress, as noted. However, while we provided evidence suggesting divergent phenotypes in mtDNA copy number and lesions following V_2_O_5_ exposure, we only included two inbred strains in this current work. Thus, future studies that include a sufficient battery of diverse inbred strains could improve our understanding of genetic susceptibility to mtDNA lesions and the potential accompanying alterations to mitochondrial dysfunction that could ultimately lead to pulmonary inflammation and pulmonary fibrosis. 

While we identified mitochondrial genomic impairments in V_2_O_5_-exposed D2 animals, we did not find any changes in nDNA lesions on the basis of strain, exposure, or time point (Figure 4). This is a conflicting finding to those in the current literature [30,32,33,34]. However, our results and the literature may be incongruent due to cell type, in vitro versus in vivo, or species-specific responses (e.g., nDNA damage and breaks occurring in human cells versus hamster cells). For example, in vitro studies of leukocytes of human cells [32,33,34,35] and Chinese hamster lung fibroblast cells [36] show that V_2_O_5_ exposure causes irreparable nDNA lesions. Thus, future investigations are likely needed to reconcile these differences and better understand the influence of V_2_O_5_ exposure on nDNA lesions. 

In summary, we acknowledge that our findings in this pilot investigation have limitations in their application. One major limitation of this current work is the low sample size and number of inbred mouse strains used. A larger sample size—as is required in genome-wide association studies using inbred mice—is needed to identify the potential mitochondrial genetic determinants that associate and interact with the differential IPF fibrosing and inflammatory phenotypes that result in IPF following V_2_O_5_ exposure. Additionally, while correlational analyses between heteroplasmy and mitochondrial copy number might reveal potential links between these factors that further our understanding of V_2_O_5_-induced IPF, these were deemed inappropriate due to the low sample size and inherent variability of heteroplasmy. Notwithstanding, this is the first study to assess the role of the mitochondrial genome on the divergent phenotypes presented by the resistant B6 and responsive D2 strains to V_2_O_5_. Further, this work provides us with preliminary data to support future work to sequence the mitochondrial genomes of a diverse inbred mouse panel to understand how the mitochondrial genome may relate to these divergent phenotypes.

## 4. Materials and Methods

### 4.1. Animals

We purchased B6 and D2 male mice (5 to 7 weeks of age) from Jackson Laboratories (Bar Harbour, ME, USA). This protocol conformed to the standards of humane animal care and was approved by the National Institute of Environmental Health Sciences Animal Care and Use Committee. We extracted lung tissue samples from randomized groups of B6 (*n* = 14 mice) and D2 (*n* = 13 mice) animals at 14- and 112-days post-PBS or V_2_O_5_ exposure (*n* = 3–4 mice/time point/treatment group) and performed mtDNA sequencing to analyze the mitochondrial genome characteristics of the lung tissue. 

### 4.2. V_2_O_5_ Exposure

We lightly anesthetized mice with isoflurane gas and suspended approximately 50 μL doses of V_2_O_5_ particles (Sigma-Aldrich, Milwaukee, WI, USA) in PBS (4 mg/kg V_2_O_5_) or PBS alone by oropharyngeal aspiration [10] on study days 0 and 7. We euthanized mice with isoflurane gas on study day 14- or 112-days post-exposure (*n* = 3 to 4 mice/dose/time point). 

### 4.3. Mitochondrial Ultra-Deep DNA Sequencing

We employed a long-range PCR (SequalPrep Long PCR Kit, Life Technologies, Grand Island, NY, USA) method to amplify the complete mitochondrial genome. First, we used the following overlapping primers to amplify the mtDNA in two halves: Set 1 (10 Kb amplicon) with forward (3301, GCC AGC CTG ACC CAT AGC CAT AAT AT) and reverse (13367, GAG AGA TTT TAT GGG TGT AAT GCG G); Set 2 (7.5 Kb amplicon) with forward (12791, TCC CAC TCC TAA ACA CAT CC) and reverse (3880, TTT ATG GGG TGA TGT GAG CC). For primer set 1, we used the following PCR conditions: 94 °C for 2 min; 10 cycles: 94 °C for 10 s, 57 °C for 30 s, 68 °C for 10 min; 25 cycles: 94 °C for 10 s, 57 °C for 30 s, 68 °C for 12 min; 72 °C for 5 min (Gene Amp PCR system 9700, Applied Biosystems, Foster City, CA, USA). For primer set 2, we used the following PCR conditions: 94 °C for 2 min; 10 cycles: 94 °C for 10 s, 51 °C for 30 s, 68 °C for 7.5 min; 25 cycles: 94 °C for 10 s, 51 °C for 30 s, 68 °C for 9 min; 72 °C for 5 min. Lastly, we cleaned the PCR products with the Zymo DNA Clean and Concentrator kit (Zymo, Irvine, CA, USA) and pooled the mtDNA for library preparation.

### 4.4. Nextera XT Mitochondrial DNA Library Preparation

For the mtDNA library preparation, we employed the same protocol outlined in our previously published work [10]. Briefly, we used 1 ng of mtDNA for each library preparation with the Nextera XT Sample Preparation Kit according to the manufacturer’s instructions (Illumina, San Diego, CA, USA). The fragmented mtDNA was amplified with limited-cycle PCR using a Nextera XT Index Kit. The PCR was completed using the following cycling parameters: 72 °C for 3 min, 95 °C for 30 s, 12 cycles of 95 °C for 10 s, 55 °C for 30 s, 72 °C for 30 s, a final extension of 72 °C for 5 min, and a hold at 10 °C. According to the manufacturer’s instructions, small fragments were removed from the PCR reaction by incubating the sample for 2 min with 90 µL of Agencourt AMPure XP beads (Beckman Coulter, Indianapolis, IN, USA). Cleared supernatants were transferred to a new microcentrifuge tube, and the libraries were measured using the Qubit dsDNA High-Sensitivity Kit (Thermo Fisher Scientific, Waltham, MA, USA). Libraries were sequenced on a MiSeq instrument (Illumina) using a 2 × 150 bp paired-end protocol with 20 samples per lane.

### 4.5. Mitochondrial DNA Alignment and Variant Calling

Like the mtDNA library preparation, our procedures for aligning mtDNA and call variants adhered to the guidelines we used in our previously published work [10]. As such, and as previously described, we aligned read pairs using bowtie2, version 2.0.0-beta7, to an index composed of the human mitochondrial genome acquired from GenBank in May 2015 (accession NC_012920). The alignments were performed in “—local” mode using the “—sensitive-local” preset options to allow insertions and deletions relative to the reference and clipping of ends extending beyond the edges of the artificially linearized reference sequence. Fragment lengths of up to 10 kb were allowed, as well as a single mismatch per seed alignment (-X 10000, -N 1). Variants were identified with a custom script using a method adapted from Hodgkinson et al. [37]. We determined each sample’s depth per allele per strand and allowed a minimum base and alignment quality score of 20. The depth calculation for sites adjacent to or within homopolymer runs considered only reads traversing the entire repeat. Areas with less than 1000× coverage were not considered, and variants must be observed at a 1% or higher frequency with a plus-to-minus strand coverage ratio greater than 0.1 and less than 0.9. The probability of observing each alternate allele by chance was calculated using a Poisson distribution, with an expected error rate of 0.01 for single nucleotide polymorphisms (SNPs derived from the quality score threshold) based on observations reported by Minoche et al. [38]. These *p*-values were adjusted for multiple testing using the Benjamini and Hochberg FDR method, and a significance threshold of 0.05 was applied. The *p*-values were calculated at each position within the mitochondrial genome based on the read counts supporting variants and reference alleles. The global null hypothesis was that no variants were present, and all reads supporting alternate alleles resulted from PCR and sequencing errors. The Benjamini and Hochberg FDR method for multiple testing correction was applied due to the large number of *p*-values calculated, as high as 49,707 for a single sample, if no positions or alternate alleles were filtered. Alternative amino acids were identified for all SNPs based on annotations and protein sequences downloaded from Ensembl in June 2015.

### 4.6. Identification of Informative Mitochondrial DNA Variants and Heteroplasmy

We defined ‘informative positions’ as positions in the mitochondrial genome (16,569 total positions) where mice in each group differed from other groups according to age, model, and/or exposure. In addition, we defined a position with heteroplasmy by the following conditions: A position with at least 1000 read sequencing depth where the alternate allele had one percent or higher occurrence than the reference allele. 

### 4.7. Mouse DNA Lesions (Mitochondrial and Nuclear) and Mitochondrial Copy Number Assays

DNA was extracted from the lungs of mice (*n* = 60) using the DNeasy Blood & Tissue Kit (Qiagen, Carlsbad, CA) by the manufacturer’s instructions. Isolated DNA was quantified with the Qubit™ dsDNA HS Assay Kit on a Qubit Analyzer (Invitrogen, Life Technologies, Grand Island, NY, USA) in triplicate for accurate quantification. All samples were diluted to 3.0 ng DNA/µL in Tris-EDTA (TE) buffer (Promega; Madison, WI, USA). We then used a quantitative polymerase chain reaction (QPCR) protocol developed by Santos [21] and Furda et al. [22] that used gene-specific primers to assess for DNA lesions (mitochondrial and nuclear) and mtDNA copy number, as provided in their protocols for mice. 

Briefly, to prepare the PCRs, the GeneAmp XL PCR kit was used with the following: 15 ng of total genomic DNA, 1X buffer, 100 ng/μL final concentration of BSA, 200 μM final concentration of dNTPs, 20 pmol of each primer, 1.3 mM final concentration of magnesium, and nuclease-free water to a total volume of 45 μL. For the mtDNA assays only (mtDNA lesions and copy number), we linearized the mtDNA for adequate amplification of each sample (40 µL total for triplicate analysis) using a restriction enzyme digest (New England BioLabs Inc., Ipswich, MA, USA) in the following conditions: Two hours at 37 °C; 3.3 µL of nuclease-free water, 5 µL of 1X CutSmart^®^ Buffer, 0.5 µL of 10X Bovine Serum Albumin, and 1.25 µL of the HaeII enzyme. Each PCR was started with a “hot start” for two minutes at 75 °C before adding the enzyme for the short- (SMITO; FailSafe Taq Polymerase; Lucigen, Middleton, WI, USA) and long-mitochondrial gene fragment (LMITO; LongAmp Taq DNA Polymerase; New England BioLabs Inc., Ipswich, MA, USA), and the nuclear gene (β-Pol; LongAmp Taq DNA Polymerase; New England BioLabs Inc., Ipswich, MA, USA). The QPCR conditions for each primer are provided in the protocol by Santos [21] and Furda et al. [22]. Experimental controls included a non-damaged control (3.0 ng/µL), a non-damaged 50% control (1.5 ng/µL), and no DNA (TE buffer). In addition, a PCR tube containing 1X TE instead of DNA (“no template” control) and a PCR tube containing 50% DNA amount (DNA diluted 1:1 first) were used to ensure optimization of the PCR cycles. A fluorescence reading was then obtained using the FL600 Microplate Fluorescence (Bio-Tek; Winooski, VT, USA), and DNA lesions and mitochondrial DNA copy numbers were calculated as described in the protocol by Santos [21] and Furda et al. [22]. 

### 4.8. Statistical Analyses

All data are presented as means ± standard error (SE) for each group. A three-way ANOVA was employed with treatment (V_2_O_5_, PBS), strain (D2, B6), and time (14 days, 112 days) as the factors for measures of mean percent heteroplasmy frequency at a heteroplasmic position (see methods section regarding our classification of heteroplasmic positions). A three-way ANOVA was also employed with treatment (V_2_O_5_, PBS), strain (D2, B6), and time (14 days, 112 days) as the factors for measures of mtDNA copy number and DNA damage (nuclear, mitochondrial). Tukey’s *post hoc* testing was used to determine statistical differences between groups, and *p* < 0.05 was accepted as statistically significant. JMP software (v. 16, accessed on 31 July 2023, SAS Institute, Cary, NC, USA) was used for all statistical analyses.

## 5. Conclusions

In conclusion, while we only assessed mitochondrial genome characteristics in two inbred mouse strains, we were able to demonstrate damage to the mitochondrial genome that was evident by increased mtDNA lesions and lowered copy numbers in susceptible mice at 112 days post-V_2_O_5_ exposure. The divergent responses between B6 and D2 strains signal the need for future research into the potential role of heteroplasmy, mtDNA copy number, and mtDNA damage on V_2_O_5_-induced fibrosis and inflammation. While our data are preliminary, they show a robust influence of both the mitochondrial and nuclear [10] genomes on pulmonary fibrosis susceptibility. Future research that includes an extensive battery of inbred mouse strains could elucidate additional mitochondrial and nuclear associations and interactions with the IPF phenotype. 

## Figures and Tables

**Figure 1 ijms-24-14507-f001:**
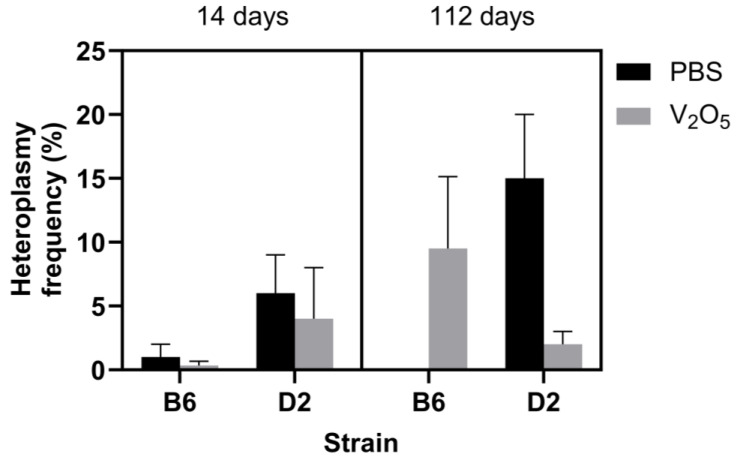
Effect of V_2_O_5_ exposure on the mean percent heteroplasmy frequency across heteroplasmic loci in two inbred strains at 14- and 112-days post-exposure. Mean ± S.E.M. presented (*n* = 3–4/group). B6, C57BL/6J; D2, DBA/2J; PBS, phosphate-buffered saline; V_2_O_5_, vanadium pentoxide, mean percent heteroplasmy frequency, mean percent heteroplasmy frequency across heteroplasmic loci; heteroplasmic loci, a position in the mitochondrial genome where the amino acid sequence changes ≥ 1% in a sample population. A Three-way ANOVA was employed with strain (B6, D2), exposure (PBS, V_2_O_5_), and time (14-, 112-days post-exposure) as the factors.

**Figure 2 ijms-24-14507-f002:**
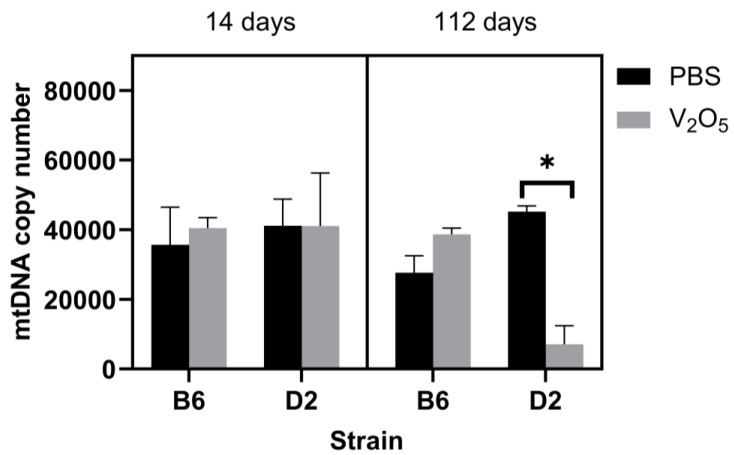
Effect of V_2_O_5_ exposure on mitochondrial DNA copies in two inbred strains at 14- and 112-days post-exposure. Mitochondrial DNA copy number (mtDNA copies/3 ng/µL) was determined in B6 and D2 mice exposed to vanadium pentoxide (V_2_O_5_) or phosphate-buffered saline (PBS) at 14- and 112-days post-exposure. A Three-way ANOVA was employed with strain (B6, D2), exposure (PBS, V_2_O_5_), and time (14-, 112-days post-exposure) as the factors. * Indicates statistical significance where there were main effects. (*p* < 0.05).

**Figure 3 ijms-24-14507-f003:**
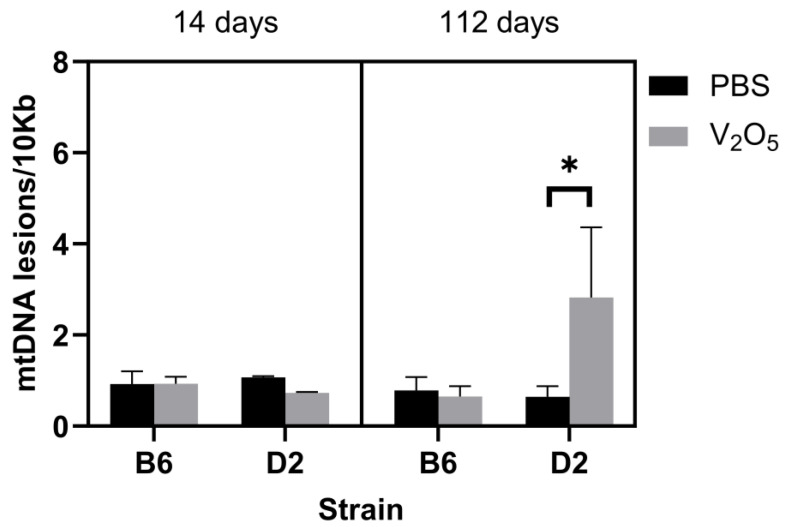
Effect of V_2_O_5_ exposure on mitochondrial DNA lesions in two inbred strains at 14- and 112-days post-exposure. mtDNA lesions (mtDNA lesions/10Kb) were determined in B6 and D2 mice exposed to vanadium pentoxide (V_2_O_5_) or phosphate-buffered saline (PBS) at 14- and 112-days post-exposure. A Three-way ANOVA was employed with strain (B6, D2), exposure (PBS, V_2_O_5_), and time (14-, 112-days post-exposure) as the factors. * Indicates statistical significance where there were main effects. (*p* < 0.05).

**Figure 4 ijms-24-14507-f004:**
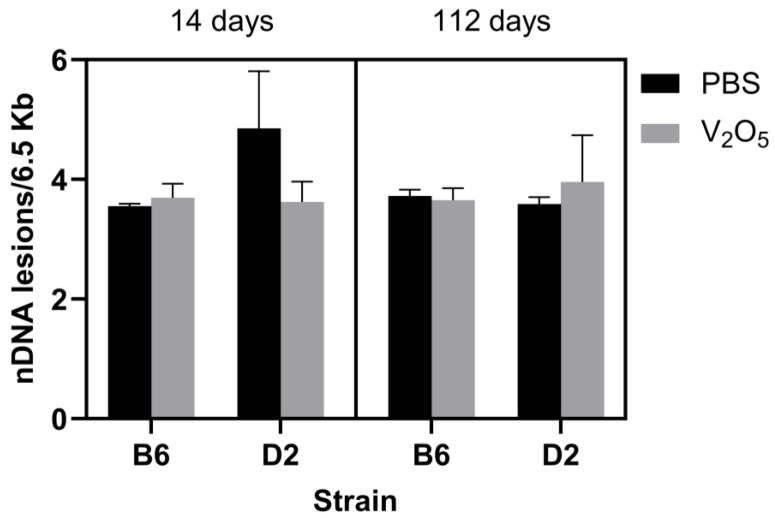
Effect of V_2_O_5_ exposure on nuclear DNA lesions in two inbred strains at 14- and 112-days post-exposure. Nuclear DNA lesions (nDNA lesions/6.5Kb) were determined in B6 and D2 mice exposed to vanadium pentoxide (V_2_O_5_) or phosphate-buffered saline (PBS) at 14- and 112-days post-exposure. Mean ± S.E.M. presented (*n* = 3–4/group). A Three-way ANOVA was employed with strain (B6, D2), exposure (PBS, V_2_O_5_), and time (14-, 112-days post-exposure) as the factors.

**Table 1 ijms-24-14507-t001:** Heteroplasmic Sites and frequency in two inbred mouse strains at 14- and 112-days post-V_2_O_5_ exposure. Heteroplasmic site * Indicates a nonsynonymous mtDNA variant at a given position in the mitochondrial genome. B6, C57BL/6J; D2, DBA/2J; PBS, phosphate-buffered saline; V_2_O_5_, vanadium pentoxide; heteroplasmic site, a position in the mitochondrial genome where the amino acid sequence changes ≥ 1% in a sample population; heteroplasmy frequency, combined percent heteroplasmy frequency in the specified group according to strain, exposure, and time. *n* = 3–4/group.

Strain	Exposure	Time Point	Position	Reference Base	Gene Name	Heteroplasmy Frequency
D2	PBS	14	4405	A	*mt-Nd2*	8.9%
D2	PBS	14	11,879	T	*mt-Nd5*	2.8%
D2	V_2_O_5_	14	11,879	T	*mt-Nd5*	8.1%
B6	PBS	14	13,052	T	*mt-Nd5*	3.0%
B6	V_2_O_5_	14	14,992 *	C	*mt-Cytb*	1.3%
D2	V_2_O_5_	112	1054	G	*mt-Tv*	2.7%
D2	PBS	112	2781 *	G	*mt-Nd1*	10.3%
B6	V_2_O_5_	112	5228	G	*mt-Tc*	7.2%
B6	V_2_O_5_	112	9214 *	T	*mt-Co3*	4.1%
B6	V_2_O_5_	112	9461	T	*mt-Nd3*	26.0%
D2	PBS	112	11,879	T	*mt-Nd5*	20.2%
B6	V_2_O_5_	112	13,270 *	A	*mt-Nd5*	1.1%
D2	V_2_O_5_	112	15,446	A	*D-loop*	1.4%

* Nonsynonymous mtDNA variant.

## Data Availability

Data are contained within the article.

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
