# Peer review of "Vanadium Pentoxide Exposure Causes Strain-Dependent Changes in Mitochondrial DNA Heteroplasmy, Copy Number, and Lesions, but Not Nuclear DNA Lesions"

_ijms, 2023, doi:10.3390/ijms241914507_

Round 1

Reviewer 1 Report

Dear authors and editor, 

This article, by Heather L Vellers et al, presents data on V2O5 affecting mitochondrial DNA. Although it is well written and written, I found it quite difficult to follow its structure and to understand the implications. I think the results section should be written in a more pedagogical way to reach a wider audience. In this respect, it is worth noting that the article is supported by graphs (4) of columns and error bars. In my opinion, it is an article that falls rather short and could be used to go deeper into the subject. Even so, in this format, I see it as publishable. 

I have only a couple of additional comments:

1) In all graphs, please add all points used. Don't just leave the graphs as a bar and your error; add the points. This will help the scientific community to assess the validity of the graphs. 

2) On line 147 (I assume it is a typo) a data is given with the following error: 24.066 +- 24.044. Is this correct? 

Reviewer 2 Report

Dear Nick Dobson and collagues,

I read your manuscript on "Vanadium pentoxide exposure causes strain-dependent changes in mitochondrial DNA heteroplasmy, copy number, and lesions, but not nuclear DNA lesions” with interests.

Here are my comments:

The names mouse strains should be the same throughout the text. Sometimes they appear as C57BL/6 or B6 and DBA/2J or D2. I think B6 and D2 are fine. I would like to suggest to use B6R and D2S to indicate resistance and sensitivity throughout the manuscript.

Although the strains are described in the Introduction, it would be good to repeat the description at the beginning of the Result section since many readers skip the introduction.

L93: say that mt-Nd3 encodes Mitochondrially Encoded NADH Dehydrogenase

L121 at the end of the section starting “At 112 days.. is a conclusion missing. I would also tell the reader briefly how heteroplasmy was measured, why a 1% cut-off threshold was used and why it is important to know about it at the beginning of section 2.1 Generally speaking, all Result sections should tell the reader why an experiment was done, how and what the key observations are. The manuscript is very much written for a narrow audience. It would benefit from a stronger narrative guiding the reader through the story.

Table 1 should be moved to Materials & Methods. The content of the table is difficult to understand

In section 2.2 in line 148 a conclusion is missing at the end of the paragrapg starting “For the mtDNA copy number, …”

In line 156 it needs to be explained what is defined as a lesion in the mtDNA (i.e. deletions, point mutations, frame shifts, inversions, duplications). If your data allow for it, please tell the reader about the frequencies of different lesions after V2O5 exposure.

Please refer to the relevant display items also in the Discussion.

Line 216 please explain why you think that a decrease in heteroplasmy corresponds to fibrosing and inflammation.

Line 224 does the increase in mtDNA lesions cause a drop in the copy number due to replication mtDNA problems?

The Discussion should also address in more detail the observation of 26% heteroplasmy in mt-Nd3 upon V2O5 treatment and the 20,2% in mt-Nd5 upon PBS treatment. Why does PBS have an impact at all? (also in the D2 strain at 112 days in Figure 1)? This needs to be addressed in the discussion.

Line 266: the link between the mutations in mt-Nd5, mt-Nd3 and nuclear gene expression is unclear. I think the discussion should not be cut short here by saying it is beyond the scope of this work.

Please tell the reader what the mt-genes encode (e.g. mtNd5 encodes Mitochondrially Encoded NADH Dehydrogenase 5).
